# New Strategies to Increase the Abiotic Stress Tolerance in Woody Ornamental Plants in Mediterranean Climate

**DOI:** 10.3390/plants12102022

**Published:** 2023-05-18

**Authors:** Luca Leotta, Stefania Toscano, Antonio Ferrante, Daniela Romano, Alessandra Francini

**Affiliations:** 1Department of Agriculture, Food and Environment, Università degli Studi di Catania, 95131 Catania, Italy; luca.leotta@phd.unict.it; 2Department of Science Veterinary, Università degli Studi di Messina, 98168 Messina, Italy; stefania.toscano@unime.it; 3Department of Agricultural and Environmental Sciences—Production, Landscape, Agroenergy, Università degli Studi di Milano, 20133 Milan, Italy; antonio.ferrante@unimi.it; 4Centro di Ricerca in Produzioni Vegetali, Scuola Superiore Sant’Anna Pisa, 56127 Pisa, Italy; alessandra.francini@santannapisa.it

**Keywords:** urban environment, shrubs, green areas, landscape, abiotic stress, pollution, plant choice, biostimulants

## Abstract

The native flora of different Mediterranean countries, often woody species, was widely recognized for its ornamental potential. The shrubs, in particular, are a typology of plants very widespread in the Mediterranean environment and constituent the ‘Macchia’, the typical vegetation of this ecosystem. The use of native shrubs for the realization of ornamental green areas has been recently examined for their adaptability to abiotic stress. Abiotic stresses, in fact, are the major limiting growth factor in urban and peri-urban areas. The identification and use of tolerant ornamental species allow the reduction of management costs and preserve the aesthetical value of green areas. Tolerance to drought stress, for instance, in the Mediterranean climate can improve the ecosystem services of these plants in the urban environment. In this review, the possibility to early individuate different plant species’ mechanisms to tolerate or avoid the stresses is analysed, as well as the possibility to increase abiotic stress tolerance through genetic and agronomic strategies. The exploration of wild or spontaneous species can be a good source for selecting tolerant plants to be used as ornamental plants in urban areas. Among agronomic strategies, biostimulants, mulching, and plant combination can provide a useful solution to counteract abiotic stress in the urban environment.

## 1. Introduction

Woody plants, trees or shrubs, represent the most common plants in many natural and semi-natural environments [1]. Almost all these plants have characteristics, such as being perennial or the same structure of ramifications, that allow their use for ornamental purposes [2]. All ‘woody ornamental plants’ can be used in gardens and landscaping, thanks to the presence of flowers, more or less showy, of different colours, the colour and morphology of leaves, and the shape of the plant (height, shape, and width) [2]. The species of woody ornamental plants belong to numerous botanical families and genera; within a single species, there are numerous cultivars, expressing enormous variability [3]. An indication of the wide biological diversity of woody plants used for horticultural purposes is the “List of Names of Woody Plants” by Naktuinbouw, which contains the ‘preferred botanical names and common synonyms and trade names of almost 45,000 woody nursery plants’ [4]. The new edition (2021–2025) contains more than 8200 new names of woody plants. Shrubs, in particular, which are the plant typology on which our attention is focused in this review, are a category of woody plants widespread above all in the Mediterranean Basin, where they form the so-called in Italian ‘Macchia’, i.e., the vegetation typical of this environment.

The diffusion of this plant typology in the Mediterranean regions is justified by its adaptability to stressful climatic conditions, characterized by hot summers and low rainfall, which determine a long and droughty summer [5]. Climatic conditions of the Mediterranean environment are also found in four other regions of the Earth (California, Chile, South Africa, and some areas of Australia). The frequency of abiotic stresses, and in particular water and saline stresses due to the poor quality of the water, has selected the plants of all these environments, leading to the convergence of the morphophysiological traits of the various species, determining that all the plant communities of the Mediterranean climate are dominated by sclerophyllous evergreen shrubs. In the Mediterranean regions, water stress and poor water quality (high salt content) are among the main problems hindering the use of ornamental plants. Global climate change will certainly accentuate problems associated with water deficits and salt levels, especially in urban areas [6]. The possibility of the use of native Mediterranean shrub species can be a solution for drought [7] and saline stress [8]. It is of interest to increase the sustainability of the landscape. Native Mediterranean shrubs are able to adapt to conditions of accentuated drought, which is one of the most important factors influencing plant survival and species distribution [9]. Many woody ornamental plants and native Mediterranean shrubs are particularly suitable to use in landscape planning. These plants, in addition to their high aesthetic value, are characterized by wide biodiversity. Indeed, the Mediterranean Basin is one of the Earth’s areas with the greatest biodiversity, as it hosts 10% of the world’s higher plants in an area that is just 1.6% of the Earth’s surface [10]. Of the approximately 25,000 species recorded in the Mediterranean area, half are endemic to the region [11]. Hotspots represent about 22% of the total area of the Mediterranean Basin and host about 5500 restricted endemic species [12].

Thanks to their particular morpho-physiological characteristics, woody plants are very suitable to be used for ornamental purposes. Since shrubs are characterised, as Givnish [13] recalled, by a high number of active meristems, which are potential sites for stem regeneration, they are able to tolerate abiotic stresses more than trees. It is no coincidence, in fact, that shrubs are associated with degraded environments, where abiotic stresses are very frequent, for these reasons. They are plant species suitable for low-maintenance green infrastructures. Shrubs, perennial plants with numerous branches that branch out at or near the ground [1], are widespread in the different biomes of the Earth; their tolerance to numerous stresses allows them to ensure numerous ecosystem services [14]. These plant species are able to control temperatures, stabilize the soil, and ensure the water balance of the ecosystem, absorption, and carbon storage. The capacity to assure ecosystem services is crucial in the choice of ornamental plants for sustainable green infrastructures. From an ornamental point of view, shrubs are characterised by different features (high number of twigs, which influences their growth pattern; pulvinate shapes, which reduces their transpiration and improves the visual qualities of the green infrastructures; leaf characteristics; different blooming periods; and colours of flowers) that add interest and variety to the landscape [15].

## 2. Methodology and Literature Research

According to the objective of this review, i.e., to analyse the strategy and improve the tolerance of ornamental shrubs to abiotic stresses that are commonly frequent in urban areas, the research has been searched in the most important scientific databases such as Scopus and Web of Science. The published research papers were selected considering their impact and keywords that matched with the aim of our review. Recent works with robust statistical data analysis were considered, especially in the last 10 years. The main criterions were the ornamental value of the plants adopted and the role of human intervention in man-designed green infrastructures (gardens and urban parks). Certainly, it was taken into consideration plants that have essential ecological functions, but attention, and therefore the exclusion of some articles, was focused on agronomic and horticultural aspects, including the choice of species, which can improve tolerance to stress and therefore maximize the ecosystem services of the plants introduced by man into the city for ornamental and fruition purposes.

## 3. Ornamental Shrubby Plants in the Urban Environment

Urbanisation significantly modifies the physical environment, biological components, and ecosystem processes of cities [16]. Woody plants, trees, and shrubs become essential components of urban green infrastructures for their numerous ecosystem services and, in particular, for the reduction of pollution [14]. Another aspect linked to urban environment quality is heat island reduction, determined by green areas and vegetation, particularly important in relation to increasing global warming. Plants, thanks to their shade [17] and transpiration, are able to improve urban conditions for the inhabitants but also for tourists who favour areas with green infrastructure to spend their time outdoors [18,19]. It is well-known that urban areas have temperatures 5–7 °C higher than rural areas. Mitigation of the high temperatures is obtained by the transpiration of plants, and the efficiency depends on water availability. Ornamental shrubs are able to maintain environmental quality and offer pleasant landscape effects, even if urbanisation, with its environmental changes, exerts negative effects on plants, which mainly affect the characteristics of the leaves [20]. At the same time, the variations that, due to the stressful effects of urbanisation, are observed in plants, such as leaf thickness, unit leaf area, specific leaf area, etc., can be used as indicators of the urban environment quality [20].

## 4. Mechanism of Tolerance and/or Resistance of Ornamental Shrubs to Abiotic Stress

Abiotic stresses are the major limiting growth factor in urban and peri-urban areas. The identification and use of tolerant ornamental species allow the reduction of management costs and preserve the aesthetical value of green areas. Urban environments can be subjected to more stressful conditions than rural areas [14].

Drought tolerance is most important for agricultural production, as most of the plants are obtained in Mediterranean, semi-arid, and tropical regions [21]. Soil water limitation during drought affects evaporation, evapotranspiration, and ultimately, precipitation [22]. Plants have developed various adaptive strategies to cope with drought stress [23]. Plants adapt to drought in several ways, such as drought escape, tolerance, and avoidance mechanisms [24]. Perennials especially rely on drought tolerance [25], which can be achieved through morphological adaptations of roots, stems, and leaves. Tolerant plants have a high-water potential with higher water uptake or physiological adaptations through the reduction of transpiration [26]. Plant adaptation varies among plants depending on species, genotype, phenological development, or organ type (leaves) [27]. Optimisation of carbon assimilation with minimisation of water loss, i.e., improvement of intrinsic water use efficiency, has been described as an adaptive trait for plants that are exposed to severe drought, like, for example, the Mediterranean woody plant species [28,29].

Woody species are particularly important in this context, due to their longevity and the possibility of studying long-term adaptation mechanisms. To understand the tolerance level, many physiological traits such as the measurement of leaf water potential before sunrise and at midday, photosynthetic rate, stomatal conductance, transpiration rate, and intercellular carbon dioxide concentration were analysed. Biochemical characteristics, such as ascorbic acid, glutathione, chlorophyll content, tocopherols, amino acids, carotenoids, and soluble sugar, have also been used to control the tolerance level of plants to drought stress [30]. It has been observed that species that can retain a greater quantity of water and therefore lose less through the stomata are more tolerant to drought [31]. As reported by Galmes et al. [32], shrubs have a better ability to regulate transpiration than herbaceous plants [30].

Many of the favourable characteristics for resisting drought are present in shrubs; it is not a coincidence that in semi-arid environments, most of the plants are sclerophyllous evergreen shrubs, or deciduous or seasonally dimorphic shrubs, which possess the main adaptive approaches of perennial species to drought stress [33]. The main role of shrubs in semi-arid ecosystems lies in the fact that these plants can grow under conditions of environmental stress where trees cannot survive [34,35]. Some perennial species, such as *Euphorbia dendroides* L., a Mediterranean shrub, keep their leaves during the winter and/or spring and drop them with the onset of the hot season.

The use of Mediterranean shrubs for revegetation in semi-arid areas has increased due to their ability to adapt to severe drought conditions, which is considered one of the most important factors influencing plant survival and species distribution [36]. In the case of Mediterranean evergreens, leaves undergo several drought events that can further hinder photosynthetic capacity [37,38]. One of their most distinctive characteristics is a higher water use efficiency (WUE) at the leaf level, due to the reduced stomatal conductance but higher carboxylation capacity of Rubisco compared to evergreens of other biomes [39]. Hence, stomatal and mesophyll diffusion constraints are the most important factors limiting photosynthesis in evergreens [40]. However, Mediterranean sclerophylls are able to sustain positive CO_2_ assimilation rates at relatively low leaf water potentials compared to Mediterranean deciduous species [39]. This increased drought tolerance has been partly attributed to the robustness of sclerophyll leaves [41], which tend to sustain shrinkage and collapse, thus preventing negative effects on photosynthesis and water transport [38,42]. Beyond the response mechanisms to drought stress, ornamental plants used in landscaping must ensure an aesthetic value that can be influenced by a reduction in the number of flowers, an excessive decrease in plant growth, and a worsening of foliage quality [30]. The analysis of the mechanisms adopted by different species to overcome drought stress and reduce water loss could allow the identification of the most tolerant species to be used in arid and semi-arid environments, thus increasing the sustainability of ornamental green infrastructures (Table 1).

Salt stress is another important abiotic stress that ornamental plants and shrubs in landscaping can be exposed to. There are not numerous studies on the effects of saline stress [47]. Salinity can affect the growth of ornamental shrubs by reducing leaf growth and expansion due to osmotic effects or by toxicity due to the high concentration of Na^+^ and Cl^−^ in saline water [48]. In ornamental plants, the aesthetic value can be compromised by salinity inducing leaf necrosis or abscission [49,50]. In many ornamental species, salinity usually induces dry shoot biomass and leaf surface. Morphological adaptations such as resinous buds, and waxy leaves and stems in tolerant species allow woody plants to cope with salinity stress. The salt exclusion mechanisms are represented by smooth twigs, sunken buds, and low surface area to volume ratios (as occurs, for example, in pine needles) [51,52].

Exposure to salt can affect plant metabolism through an osmotic effect, causing water deficit, or through a specific ion effect, causing excessive ion accumulation [53]. Under saline conditions, plants must activate various physiological and biochemical mechanisms to cope with saline stress, which include water relationships, photosynthesis rate, hormonal profiles, toxic ion distribution, antioxidant metabolism, and soil response [54]. In particular, the changes in leaf tissue cell walls and factors limiting photosynthesis under these conditions and their possible interactions with leaf tissue damage are not well understood [29]. Plants that have some degree of tolerance to salinity may show quality reductions when exposed to this stress, and this is an important factor in the selection of ornamental plants for use in gardens and landscaping [55].

The ionic composition of irrigation water can influence the response of shrubs and trees to salt stress. Chloride salts appear to be more harmful than SO_4_^2−^ salts, and Mg^2+^ associated with Cl^−^ is more harmful than Na^+^ with Cl^−^ [56]. Among many salinity tolerance mechanisms [57], the ability to limit the entry of saline ions through the roots and to limit the transport of Na^+^ and/or Cl^−^ to the aerial parts, retaining these ions in the root and in the lower part of the stem, is one of the most important characteristics associated with salt tolerance [58]. Species that maintain acceptable growth rates under saline conditions have effective mechanisms for excluding Na^+^ and Cl^−^ from roots or leaves, thus maintaining good aesthetics and are ideal for landscaping. The low reduction and absence of symptoms of salt damage in *Eugenia myrtifolia* L. was associated not only with the root storage of Na^+^ and Cl^−^, but also with their limited uptake with increasing salinity [59]. An important aspect of salt tolerance is related to a plant’s ability to compartmentalise toxic ions, such as Na^+^ and Cl^−^, in roots or stems [60,61].

The response of plants to salinity depends not only on the intensity of the salt treatment but also on the time of exposure to the salt treatment [62]. These aspects are of primary importance, especially in the Mediterranean area when saline water is used for irrigation of perennial species, such as woody plants, as the interaction between intensity and duration of exposure to salt will determine physiological and molecular changes. At nursery level, the selection of plants tolerant to salinity stress can be carried out by the evaluation of plants’ responses to salinity treatments (Table 2).

In urban areas, hypoxia is an abiotic stress that ornamental plants can be often exposed to in compacted soil. Compaction is determined by physical degradation, which reduces the volume of a given mass of soil and decreases porosity. This promotes the formation of urban flooding [66]. An excess of water is usually considered to be deleterious to plant health and growth, and total submergence rapidly kills most plant species. Hypoxia/anoxia conditions restrict processes such as plant respiration and water and nutrient absorption [67,68,69]. The most common symptoms in the aerial part of a plant under hypoxia/anoxia conditions include leaf curling (epinasty) and stem twisting, leaf chlorosis and wilting, marginal browning of the leaf and shedding/defoliation, as well as fruit drop. The physiological consequences of hypoxia are a decrease in stomatal conductance [70] and a reduction of water potential [71]. In woody plants, waterlogging tolerance responses are associated with hypertrophied lenticels, new adventitious roots, and aerenchyma development [72]. These morphological and anatomical changes depend on the intensity, duration, and timing of the flooding cycle [68]. The presence of hypertrophied lenticels is a common anatomical change observed in many woody species [73]. The development of hypertrophied lenticels is supposed to simplify the downward diffusion of O_2_ as well as the potential discharge of compounds produced in the roots as by-products of anaerobic metabolism [74].

Oxygen depletion is one of the most important events during flooding. The diminishing in gas diffusion to the root environment as a result of the presence of excessive water in the soil or deprived aeration in soilless cultures, accompanied by reduction of available oxygen by aerobic processes (i.e., root and microbial respiration), will deprive the rhizosphere of available O_2_.

A flood-tolerant plant can overcome the adverse effects of flooding through numerous morphological modifications, such as hyponasty (upward bending of leaves), improved shoot extension, aerenchyma formation, the development of barriers against radial O_2_ loss (ROL) in roots, the development of adventitious roots, leaf anatomical changes, and the formation of a gas film on leaf surfaces [75,76]. The formation of adventitious roots improves the plant’s adaptation to flooding stress, effectively transports atmospheric O_2_ into the roots, and may support or replace the primary root system [75]. Furthermore, aerenchyma occurs in adventitious roots and acts as an O_2_ transportation pathway (Table 3).

Urban environment pollution is also a source of stress in ornamental plants. Urban areas can be highly polluted by human activities. Pollution can be represented by heavy metals derived from heating systems, vehicular traffic, and industrial emissions [79]. Combustion of engines and tire emissions can represent a mobile pollutant source, while industries and heating systems represent fixed sources of pollution [14]. Around pollution sites, the concentration of heavy metals increases. Ornamental plants can have different degrees of pollution tolerance or ability to uptake and degrade them if they are organic pollutants. The use of suitable plant species can recover the visual appearance of polluted areas. The success of green area establishment depends on the tolerance of ornamental plants to the pollutant concentrations. Heavy metals are represented by different elements such as aluminium (Al), arsenic (As), cadmium (Cd), copper (Cu), chromium (Cr), lead (Pb), mercury (Hg), and zinc (Zn). The tolerance of ornamental plants to heavy metals is strictly connected with the ability of these species to exclude the toxic heavy metals from the uptake or the ability of plants to uptake and translocate heavy metals to organs that can be released, such as older leaves or even fruits [80].

The identification of tolerant ornamental plants at nursery level can be performed by exposing the interested species to increasing concentrations of heavy metals. The biochemical and physiological response of plants allows their classification to different levels of tolerance. The specific markers for the evaluation of tolerance to heavy metals can be the production and accumulation of some protection molecules such as phytochelatins. These proteins can protect the plants by removing metals from the active cell metabolism by chelation, and their biosynthesis is induced by heavy metals accumulation. The phytochelatin biosynthesis is mediated by phytochelatin synthase and starts from glutathione [81]. The activity of this enzyme is regulated by the heavy metals post-translation activation [82]. Heavy metals induce plant stress with the accumulation of free radicals and damage to the cell membrane. The most common radicals are represented by reactive oxygen species (ROS), reactive nitrogen species (RNS), or reactive sulfur species (RSS). The non-specific response can be represented by the increase of the detoxification enzymes such as those belonging to the ascorbate–glutathione cycle (Appendix A).

Free radicals are highly reactive and can damage the cell membrane and the phospholipid double layers. The damage of ROS on the cell membrane is specifically due to loss of compartment integrity and enzymes coming into contact with substrates generating products that can be responsible for several physiological disorders, compromising the visual appearance [83]. Membrane integrity and low lipid peroxidation are also good markers for the estimation of ornamental plant tolerance to heavy metal concentrations. At the nursery level, the selection of plants tolerant to heavy metals is carried out by exposing the plants to increasing doses and monitoring the lipid peroxidation, phytochelatin accumulation, and enzymatic response. The distribution of plants in the planning area must be done considering the concentration and distribution of pollution in the soil.

The shadows of buildings or tall plants in green areas can have negative effects on other plants. Therefore, the combination of different plant species such as herbaceous plants, shrubs, and woody ornamentals must be carefully considered. The visual appearance and aesthetical quality of the area depend on the health status of plants and their correct distribution. It is important to identify the correct exposure to ensure adequate light intensity. Plant distribution and combination must be carried out considering their shading tolerance. Buildings and trees can be responsible for shading and light limitations. Many ornamental plants can have a plasticity degree that allows the adaptation of plants to lower light intensities. At the nursery level, the ornamental plants can be prepared for low-light environments by progressive light reduction using black nets with a shading percentage from 50 to 90%. The intensity of shading depends on the shading in the urban area. The shade adaptation must be achieved by slow light intensity reduction [84]. Plants under shade contribute to the ornamental value through the increase of chlorophyll concentration. At the physiological level, leaves under progressive shading intensity reduce the light compensation point (Figure 1). This means that a lower amount of light is required to compensate the respiration process [85]. Ornamental plants that have good light plasticity can be used for green planning in the shaded areas inside urban and peri-urban environments. If plants are not tolerant to shade, under shading conditions, the respiration can be higher than photosynthesis with a negative sugar accumulation balance in a 24 h period. This negative balance, if prolonged, can lead to plant death. Shade plants can survive at low light availability since they have low light compensation points. Plants tolerant to shade are typical of underbrush conditions (Table 4).

High and low temperatures can induce damages that compromise the ornamental value of plants. Temperature is an important environmental parameter that can induce speciation and affect plant distribution in diverse geographical areas. Plant growth and development are tightly correlated with temperature, and many species are synchronised with the environment for foliation and flowering. Each species has an optimal range of development; the minimum and maximum temperatures must be considered in the selection of plant species to use in certain regions or geographical areas. The temperature has a direct impact on primary and secondary metabolism. In an urban context, the reduction of growth is not a problem if there is any change of visual or external quality. In fact, slow growth can reduce the cost of management due to pruning. Unfortunately, the wrong ornamental plant selection exposed to low temperature can suffer cold stress or chilling injury during winter. On the contrary, plant species sensitive to high temperature, as well as for low temperature, can also show some physiological disorders such as leaf abscission or senescence.

Cold stress can be dramatically deleterious depending on the phenological stage of plants. Deciduous ornamental plants are strongly tolerant to low temperatures during winter when they are in the dormant stage. In spring, if new vegetation appears early, eventual low temperatures can induce chilling injury. Based on temperature data recorded in recent years, it is possible to distribute plants in areas considering their sensitivity to low temperatures. The combination of ornamental species from woody trees, shrubs, and herbaceous plants can protect each other. 

High temperatures during summer can negatively affect ornamental plant performance (Table 5). The negative effect of heat stress depends on the solar radiation intensity and duration. In summer, the temperature can rapidly increase during the day, and the highest values can be observed from 12:00 to 2:00 pm. It can happen that ornamental plants can be exposed to high temperatures for 4–6 h per day during the hottest summer months. The localisation and light exposure of plants in the urban environment can mitigate or increase high temperature stress. At physiological and biochemical levels, the tolerance of plants to high temperatures depends on the transpiration rate and the thermoregulation efficiency of plants [88]. At the molecular level, the tolerance of plants to heat stress is associated to the accumulation of heat shock proteins (HSPs) and genes encoded for the detoxification of ROS [89].

## 5. Strategies to Improve the Shrub Tolerance to Abiotic Stress

The improvement of shrub tolerance can involve the identification of new ornamental species among wild or spontaneous plants in specific geographical areas. These works involve genotype characterisation or agronomic strategies such as the use of biostimulants or mulching.

### 5.1. Plant Species and/or Cultivar Choice

Due to their characteristics, woody plants, trees, and shrubs can ensure numerous ecosystem services. Plants in cities provide valuable ecosystem services, which can improve quality of life, but they also face numerous stresses, such as heat, salt, drought, extreme winds, and pests, which can reduce or cancel these benefits [94]. In order to ensure ecosystem services, a key factor is plant selection to obtain more stable and resilient green infrastructures. The possibility of fully utilising the benefits of green infrastructures and their ecosystem services is reduced by the lack of technical criteria for the selection of plant species in urban areas [95,96,97]. As recorded by Capotorti et al. [98], it is essential to put ’the right plant in the right place‘. There is not much research on the requirements and characteristics of ornamental plants in urban environments [99]. The lack of a clear reference framework [100,101] leads to unsuitable choices and often an increase in the maintenance costs of green areas. The selection of plant species is, in fact, influenced by subjective elements, such as the availability of plants in nurseries or personal preferences, which determines the planting of species unsuitable for the environment and which, therefore, do not ensure the desired ecosystem services.

The choice of a single species can be done for urban parks on the basis of their adaptability to environmental conditions, their aesthetic and ecological values, their reduced maintenance requirements, and the advantages they can bring (Figure 2). The selection of appropriate species of ornamental plants becomes the key factor for the creation of a sustainable green space, i.e., a space that adapts well to the environmental conditions of the place [102,103]. In this context, it is essential to identify a diversified mix of woody species well-adapted to the conditions determined by climate change. The adaptation of the plant depends, in addition to the characteristics of the genotype, on the environmental conditions and the amount of care it will receive [104]; it should also be remembered that rates of climate change can be more rapid and extreme in cities than in rural areas. Miller [105] proposed a species selection scheme that included site (i.e., environmental and cultural constraints), social factors (i.e., aesthetics, functions, and disruptions), and economic factors (i.e., planting and maintenance costs). Roloff et al. [106] focused on drought tolerance and cold hardiness as critical for the future survival of trees in a changing climate, based on the climatic conditions of the species’ places of origin.

This reasoning, although correct, must, however, take into account, in addition to the characteristics of the environments of origin and therefore the needs of the different species, the physiological plasticity of a plant, i.e., the range of habitats to which a species can adapt. It should also be taken into account that one stress can accentuate the severity of another [107]; higher temperatures, for example, increase the evapotranspiration demand and therefore drought stress, predisposing the plant to attacks by parasites [108]. In addition, salinity from recycled irrigation water or coastal flooding can adversely affect soil health and shrub growth [109]. Species with narrow tolerance ranges may be most adversely affected. Some shrubs, thanks to their plasticity, often appear more suitable than trees, which have a poor genetic ability to adapt due to their long life.

The complexity of choosing the correct plant species determines a reduction in the number of species used, considered more ‘reliable’. However, the biodiversity of urban green spaces is important as it reduces the risks deriving from pests and diseases and from climate change and therefore improves the resilience of ecosystem services ensured by green infrastructures. To manage and enhance biodiversity, Santamour’s proposed 10/20/30 ‘rule of thumb’ [110] has been widely accepted, which states that urban forests should comprise no more than 10% of any particular plant species, 20% of any genus, or 30% of any single botanical family [111].

Urban environments are capable of sustaining plant species biodiversity due to environmental and land cover heterogeneity, socioeconomic factors, and possible species introductions. Urban green spaces include many different types of habitats, from intact patches of native vegetation to highly constructed habitats such as green roofs [112]. Improving our understanding of the contribution of urban biodiversity could enable green infrastructure to deliver the ecosystem services needed to sustain an urbanising global population. A very important contribution to the biodiversity of the urban forest can be ensured by shrubs, due to their small size, which allows for the presence of numerous individuals within the green area, and their resilience to numerous biotic and abiotic stresses.

### 5.2. Agronomic Tools and Management Plans

The tolerance of ornamental plants can be induced or activated by different applications of microbic and non-microbic biostimulants [113] or other strategies that can directly or indirectly reduce abiotic stress intensities. 

#### 5.2.1. Biostimulants and Arbuscular Mycorrhizas

Landscape plants are produced in nurseries in a relatively controlled environment. Upon transplanting to the landscape, they are subjected to high levels of post-transplant stress, caused by such factors as root loss, water stress, insects and disease, and soil changes [114]. Therefore, in the first period of establishment, it is essential to minimise stress for plants with the best growing conditions possible. In recent decades, globally, high temperatures, drought, salinity, and heavy metals are considered main stresses that have negative effects on plants. In order to overcome these stress conditions, plants implement various mechanisms (morphological, physiological, biochemical); to relieve these stressful conditions, biostimulants have been widely used in recent years. The use of biostimulants in agriculture has been emphasised, which are products that contain active ingredients or organic agents free of pesticides, capable of acting, directly or indirectly, on all or part of the cultivated plants, increasing their productivity. According to EU Regulation 2019/1009 [115], biostimulants means a product stimulating plant nutrition processes, independently of the product’s nutrient content, with the sole aim of improving one or more of the following characteristics of the plant or the plant rhizosphere: (a) nutrient use efficiency; (b) tolerance to abiotic stress; (c) quality traits; (d) availability of confined nutrients in soil or rhizosphere. Biostimulants are products based on natural raw materials, such as hydrolysed proteins and amino acids from animal and plant by-products, microalgae and seaweed extracts, humic substances, plant extracts, and microorganisms [116]. Biostimulants are applied as a foliar spray and enhance plant growth; freezing, drought, and salt tolerance; photosynthetic activity; and resistance to fungi, bacteria, and virus, improving the yield and productivity of many crops. Unfortunately, new landscape plantings are often installed in settings with poor soil (e.g., heavy clay, low organic matter, and poor nutrition) and receive slight or no supplemental irrigation, which may decrease their chance of survival. Plants treated with commercial biostimulants including mycorrhizae, hydrogel, and/or biostimulants may be more resistant to such stressful conditions, necessitate less additional nutrients and irrigation, and have improved disease resistance [117]. Biostimulants reduce the need for fertilizers for the plants and increase their productivity and resistance to water stress since they act as a hormonal and nutritional increment [118].

In shrubby plants, production can be improved by biostimulant application. Hibiscus (*Hibiscus* spp.) treated with commercial biostimulants showed an increase in gas exchange with higher photosynthetic activities [119]. Hibiscus plants treated with hydrolysed substances obtained from green compost and a fraction of urban solid wastes (i.e., FORSU) showed an enhanced photosynthetic rate that turned into a higher relative growth rate and biomass accumulation under optimal growing conditions [120].

Actiwave, a biostimulant based on carbon nitrogen, was utilised in the nursery for improving the rooting of *Camellia japonica* L. cuttings because the rooting stage in this species is long and requires more than three months if no rooting promoting treatments are applied [121]. Camellia cuttings treated with Actiwave^®^, Valagro, Atessa (Chieti), Italy as a spray, showed a speeded uprooting and growth.

In a study on woody ornamental plants using *Lantana camara* L. treated with humic substances, the genetic analyses (MADS-box AGAMOUS-like) highlighted the relationship between the above substances and the activation of genes involved in plant flower and fruit development [122].

Arbuscular mycorrhizal (AMF) and humic substances are two of the seven main categories of biostimulants, which have been used separately in previous studies to improve plant nutrient uptake, growth, and development, as well as to enhance tolerance and resistance to abiotic stress and to promote soil structural stability [123,124].

On gardenia (*Gardenia jasminoides* J. Ellis), a calcifuge woody plant, the effects on the growth of two strains of *Rhizophagus irregularis* (AMF) were analysed to evaluate the possibility to reduce phosphate fertilization. Under reduced phosphate fertilization, inoculation with arbuscular mycorrhizal fungi favoured the growth of gardenia plants, especially in the high-peat substrate [125].

Many studies indicate that the symbiotic relationship between plants and the arbuscular mycorrhizal fungi (AMF) is a key factor in helping plants tolerate and/or resist abiotic stress. Yang et al. [126], in a study conducted on *Robinia pseudoacacia* L. under lead stress, showed that AMF symbiosis had enhanced the physiological and biochemical properties of this woody plant via increased ROS scavenging capacity.

Earlier rooting of photinia cuttings was observed in plants treated with the rhizobacteria *Azospirillum brasilense* [127]. In a study conducted by Loconsole et al. [128] on *Lantana camara* L. and *Abelia × grandiflora* (Rovelli ex André) Rehder, the treatments with Goteo^®^, (Goteo—Goactiv, UPL, Cesena, Italy) a commercial seaweed-based biostimulant, stimulated adventitious rooting and provided better rooting quality and shoot development of stem cuttings. Similarly, Rathore et al. [129] showed that the plants of *Glycine max* (L.) Merr. treated with seaweed extract (prepared from *Kappaphycus alvarezii*) showed beneficial effects with improved vegetative growth.

The application of biostimulators commonly results in an increased concentration of photosynthetic pigments that are strictly connected with the plant’s photosynthetic activity and carbohydrate levels [130]. 

In this regard, Augé [131] stated, also, that the positive effect of AMF relies mainly on the uptake and transport of water and on an improved uptake of nutrients, especially of available soil P and other immobile mineral nutrients, resulting in the hydration of plant tissues, sustainable physiology, and a clear promotion of growth. AMF have also been shown to regulate several plant growth-controlling processes both under normal and stressful conditions [126,132]. Arbuscular mycorrhizal (AM) symbiosis can also increase host resistance to drought stress, although the effect is not always predictable.

The positive effect of plant growth-promoting bacteria (PGPB) occurs through the activation of the 1-aminocyclopropane-1-carboxylate deaminase enzyme that reduces ethylene production and increases auxin concentration in roots [133]. Plant growth regulators (PGRs) act at very low concentrations to stimulate, inhibit, or otherwise modify plant growth. PGRs are commonly used for root induction and development in cuttings propagated in ornamental shrub nurseries. Auxins are particularly capable of stimulating simultaneous and steady root formation.

#### 5.2.2. Mulching

Mulching in urban areas has multiple functions. The mulch can be used for weed control, avoiding competition among ornamental plants and spontaneous species. The mulch can also be useful for reducing water loss from the soil and improving the water use efficiency of ornamental plants used in the green area. This barrier effect in reducing water evaporation has been studied in zinnia plants mulched with pine needles, volcanic stone (scoria), a plastic polyethylene layer, and wood chips [134]. Results showed that the water use efficiency of plants increased in plants mulched with plastic polyethylene and wood chips. Weed development was absent in plastic and wood chip mulching. The flowering periods were longer with a lower number of flowers in plants mulched with plastic polyethylene. In woody ornamental plants, such as *Tilia europaea* L. and *Aesculus × carnea* Zeyh., trees mulched with two organic mulch coarse composts derived from green material left after sifting (coarse compost) and pine bark were analysed in terms of growth or eco-physiological response [135]. The two organic mulches increased the height increment and the transpiration of plants compared to the control treated with herbicide. In a study performed using cobblestone, water permeable brick, pine bark, green waste compost, and living (turf grass) mulches on the growth of *Sophora japonica* L. in urban trees, results indicated that organic mulching improved soil fertility and physical properties, but no differences were found in the growth [136]. Similar results were observed in tea olive [*Osmanthus fragrans* (Thunb.) Siebold]. Mulching used was inorganic, such as round gravel, and organic, such as wood chips, and manila turf grass demonstrated that organic mulches improved soil organic matter [137].

#### 5.2.3. Association among Different Species

The combination of plants is an important parameter that must be carefully considered in urban areas to reduce management costs and avoid agronomic problems. The combination must be planned to have at least no negative interference among the plants at the roots and aerial parts. As for herbaceous plants, the combination of plants should be planned avoiding competition for nutrients or water at the root level. Plant density and species distribution should be done by placing plants with superficial and deep roots closer to exploit nutrients at different depths. The herbaceous plants and trees should be combined favouring the use of plants belonging to Fabaceae family that can provide nitrogen by symbiosis with *Rhizobium* that are able to fix the atmospheric nitrogen. This is important for avoiding the supply of nitrogen fertilizers and lowering management costs. Positive interaction can be also represented by the improvement of soil structure. Trees can improve water infiltration in soil with their roots, avoiding flooding, with benefits for species suffering in heavy and compact soils. In polluted soils, species that have high heavy metal uptake can reduce the concentration for sensitive species, avoiding the appearance of phytotoxicity symptoms. 

Interactions among plants belonging to different species (interspecific) and among those belonging to the same species (intraspecific) shall be investigated in ornamental plants for combination in urban environments. Antagonistic species that produce allelopathic molecules should be planted at a minimum distance to avoid negative effects on the closer plants. All antagonism situations must be avoided because they can induce ornamental value losses such as leaf yellowing and stunted growth. In particular, it has been reported that Lantana camara L. showed allelopathic effects on several species [138], especially those that belong to Liliaceae family [139]. The allelopathic substances can cause the death of trees or herbaceous species around the producing plants. Black walnut produces natural products that can induce the death of white pine (*Pinus strobus* L.) and red pine (*P. resinosa* Aiton), or black alder (*Alnus rugosa* Spreng.) and whips of white birch (*Betula* L. spp.) [140]. However, further research studies are required for planning a better combination of ornamental plants and avoiding those that have antagonist effects.

At the areal part, the canopy of plants should be independent without branch crossing that can limit air circulation with an increase of disease incidence or insect infestations. The intersection of the branches can be avoided by increasing the planting distance.

High density induces higher maintenance costs, especially those related to pruning. The combination of plants with canopies at different heights can provide several positive benefits. In shade plants, the shadow of higher plants can show dark green leaves. The canopy of higher plants can protect the shorter ones.

#### 5.2.4. Transplanting Modalities

The transplanting modalities are crucial to assure the full establishment of plants. Information about this issue is wider for re-establishment of woody species in degraded environments than in urban green areas. The season of transplanting is crucial: in the Mediterranean climate, fall transplant before the rain season helps the establishment of the plants. The inability of container-grown seedlings to develop deep and well-structured root systems rapidly after planting out [141] hampers the plant’s survival. Different tools to increase the root system’s ability to capture and transport water efficiently are adopted, like arbuscular mycorrhizal (AM) symbioses [142,143]. 

Drought stress imposed in the nursery phase can allow the seedlings to develop root biomass and branching [143]. Gilman et al. [144] analysed the effects of different volumes and frequencies of irrigation applied to the root ball, in view to understand their influence on canopy growth, plant health, survival, and attractive features, and found that irrigation frequency affected only one species (*Viburnum odorotissimum* Ker Gawl) among the shrub species analysed (*Ilex cornuta* Lindl. & Paxt. ‘Burfordii Nana’, *Pittosporum tobira* Thunb. ‘Variegata’). Irrigation every four days with three litres was able to establish shrubs in north Florida, where rainfall occurs after planting. The survival or growth was not increased if applying more water volume or irrigating more frequently. Only a slight improvement of the aesthetical value of shrubs was obtained in the first year after planting with more intense irrigation.

To promote the establishment of some native Mediterranean shrubs (*Medicago arborea* L., *Quercus coccifera* L., and *Pistacia lentiscus* L.), alpha grass (*Stipa tenacissima* L.) was adopted. The grass significantly reduced photosynthetically active radiation and soil temperature, improving shrub survival near *S. tenacissima* than in the open areas. Predawn water potentials measured before and after the summer were significantly higher in the area with the alpha grass [145].

Micro-catchment water harvesting [146] can also represent a solution in very dry climates to capture rainwater and improve soil moisture and vegetation establishment because it allows deep root development and reduces the mortality rate of shrubs [147]. The slope length useful to capture water can be inserted by the plant composition of the green infrastructure.

In order to understand the influence of irrigation and organic mulching on the survival of shrubs after transplanting, Montague et al. [148] analysed the gas exchange and growth of some shrubs (*Lagerstroemia indica* L. ‘Victor’, *Forsythia ×intermedia* Zabel ‘Lynwood’, *Spiraea ×vanhouttei* (Briot) Zabel, and *Photinia ×fraseri* Dress) placed in landscaped beds. After transplanting, the plants were irrigated twice a week, with a return of 100%, 75%, and 50% of the reference evapotranspiration (ETO). Although the plants that had used mulch and greater quantities of water had a greater stomatal conductance, all the plants—the trial took place in Texas in a place with high temperatures (>32 °C) and low rainfall (26.3 cm)—survived, and they appeared healthy throughout the growing season, fully responding to their ornamental function.

## 6. Conclusions

Abiotic stress mitigation can be achieved by a multi-action approach, such as agronomic management and suitable ornamental genotypes. In a short period, the use of biostimulants, mulching, or appropriate species combination could provide positive effects on abiotic stress mitigation, especially in urban and peri-urban areas. Beside the agronomic strategies, the selection of tolerant species against the different abiotic stresses could provide positive effect on urban environments and human welfare. Wild or underutilised species can be opportunely selected for improving or preserving the ornamental quality.

Further studies must be carried out for understanding the ornamental plant interactions and responses under abiotic stresses in urban environments.

## Figures and Tables

**Figure 1 plants-12-02022-f001:**
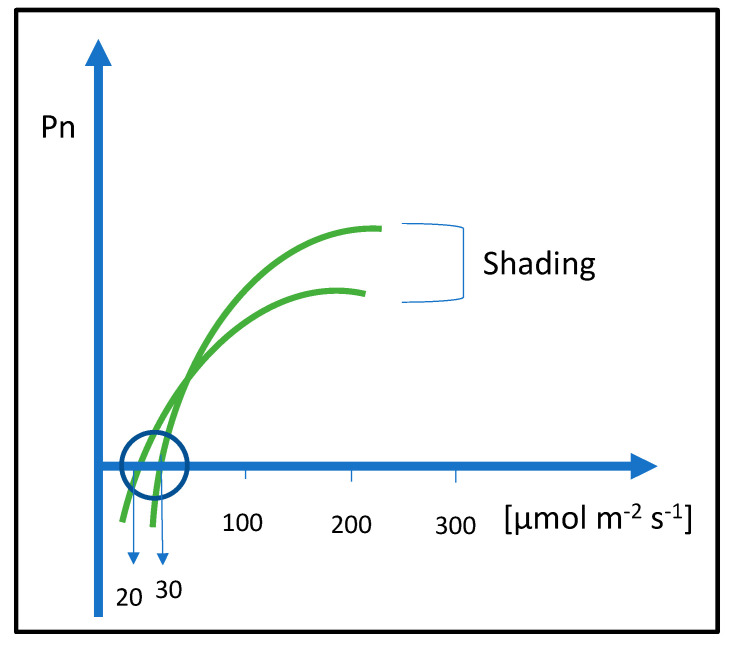
Schematic of light saturation curves and light compensation points lowered by shading treatments. Shading lowers the light compensation point from 30 to 20 µmol m^−2^ s^−1^. The lowering of light compensation point can require several days or weeks.

**Figure 2 plants-12-02022-f002:**
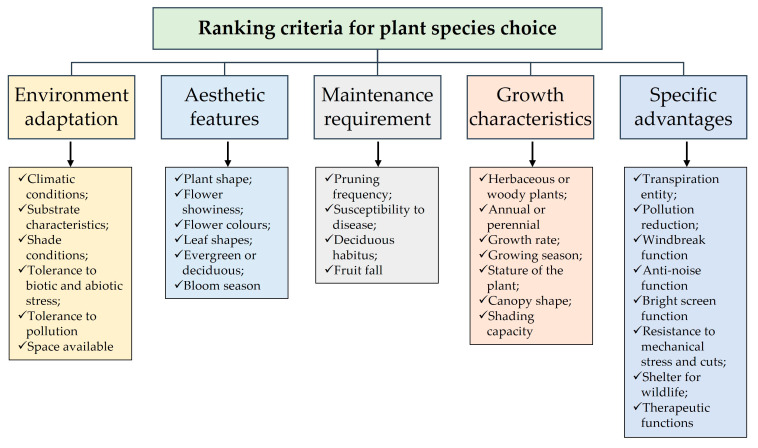
Ranking criteria for plant species choice for sustainable green areas.

**Table 1 plants-12-02022-t001:** Effect of drought on ornamental plant quality and traits associated to tolerance.

Target Organs	Stress Effects	Tolerance or Adaptation Response	References
Roots	Increase of root biomass	Increase the functional roots and architectures	[30]
Stem	Decrease the growth, elongation, diameter and biomass	Increase the lignification process (chi lo dice?)	[30,43,44]
Leaves	Reduction of size and leaf number	Increase the wax or thickness, and trichome number	[27,45]
Flowers	Reduction of flower production and longevity	Increase the flower longevity and turnover	[46]

**Table 2 plants-12-02022-t002:** Effect of salinity on ornamental plant quality and traits associated to tolerance.

Target Organs	Stress Effects	Tolerance or Adaptation Response	Reference
Roots	Increase of roots biomass	Increase the water uptake and exclusion of some toxic ions such as Na^+^ or Cl^−^	[63]
Stem	Decrease the growth and biomass	Increase the extrusion or storage	[64]
Leaves	Reduction of size, necrosis, or abscission	Increase the storage of ions in vacuole	[49]
Flowers	Reduction of flowers production and longevity	Increase the flower longevity and turnover	[65]

**Table 3 plants-12-02022-t003:** Effect of hypoxia or anoxia on ornamental plant quality and traits associated to tolerance.

Target Organs	Stress Effects	Tolerance or Adaptation Response	References
Roots	Increase the ethylene production	Cell death and in some herbaceous plants the aerenchyma formation	[77,78]
Stem	Decrease the growth and biomass	Increase the ethylene biosynthesis	[77]
Leaves	Leaf yellowing, abscission	Increase the storage of ions in vacuole	[77]
Flowers	Reduction of flower production and longevity	Increase the flower longevity	[78]

**Table 4 plants-12-02022-t004:** Effect of shadow on ornamental plant quality and traits associated to tolerance.

Target Organs	Stress Effects	Tolerance or Adaptation Response	References
Roots	Decrease of roots biomass	-	[86]
Stem	Increase of stem elongation	Increase the firmness of cell walls	[86]
Leaves	Reduction of thickness and leaf number	Increase the chlorophyll concentration and lowering the light compensation point	[86,87]
Flowers	Reduction of flower number	Increase the flower longevity and turnover	[87]

**Table 5 plants-12-02022-t005:** Effect of high or low temperature on ornamental plant quality and traits associated to tolerance.

Target Organs	Stress Effects	Tolerance or Adaptation Response	References
Roots	Decrease of roots functionality	Increase the roots uptake of ions	[90,91]
Stem	Increase or decrease of stem growth	Increase the firmness of cell walls	[92]
Leaves	Increase of thickness, leaf necrosis	reduction the chlorophyll concentration	[92]
Flowers	Reduction of flower numbers	-	[93]

## Data Availability

Not applicable.

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
