# Peer review of "New Strategies to Increase the Abiotic Stress Tolerance in Woody Ornamental Plants in Mediterranean Climate"

_plants, 2023, doi:10.3390/plants12102022_

Round 1
Reviewer 1 Report
The article on new strategies to increase abiotic stress tolerance in woody ornamental plants is intriguing and provides insightful analysis on the effects of abiotic stress. Additionally, we would like to suggest some relevant bibliographic references that could provide new directions for decreasing urban heat island, improving thermal comfort, and enhancing urban environmental conditions, such as the following: https://doi.org/10.1007/s00484-019-01678-1, https://doi.org/10.1016/j.jenvman.2022.115161, https://doi.org/10.1016/j.uclim.2022.101264.
The article is written in a very comprehensive and understandable way, which leads me to believe that the written work is of good quality. In my opinion, after reviewing these minor details, the article should be accepted. Good luck!
Author Response
Reviewer 1
Dear reviewer,
The authors would like to thank you for your comments. The manuscript has been accordingly revised. Corrections and suggestions have been implemented in the current version of the manuscript. All the modifications are highlighted in yellow in the manuscript. We hereby provide a point-by-point answer.
The authors
Comments and Suggestions for Authors
The article on new strategies to increase abiotic stress tolerance in woody ornamental plants is intriguing and provides insightful analysis on the effects of abiotic stress. Additionally, we would like to suggest some relevant bibliographic references that could provide new directions for decreasing urban heat island, improving thermal comfort, and enhancing urban environmental conditions, such as the following: https://doi.org/10.1007/s00484-019-01678-1, https://doi.org/10.1016/j.jenvman.2022.115161, https://doi.org/10.1016/j.uclim.2022.101264.
The article is written in a very comprehensive and understandable way, which leads me to believe that the written work is of good quality. In my opinion, after reviewing these minor details, the article should be accepted. Good luck!
Author Answer (A.A.): Thank you very much for your positive evaluation, comments, and suggestions. The suggested articles on heat island have been added to manuscript.
Reviewer 2 Report
General comments
The article wants to investigate, in a review form, the different stresses of plants in urban environment, such as the mechanisms of tolerance or avoiding stresses by different plant species. The paper also stresses possible agronomic strategies, bio stimulants, mulching, and plant combination which can provide a useful solution to counteract the abiotic stress in the urban environment.
Overall, the article is interesting, summarizing relevant solutions, which can be applied for improving tolerance, however, some relevant changes will improve the interest and originality of the paper.
In particular, the first part (in particular Paragraph 3 and all the following subsections until paragraph 4.1) should be highly reduced since many of such information can be found in general texts of plant ecology, and it will much more coherent with the title.
The structure doesn’t follow the typical structure of a research or review paper:
the aims are not declared (they are indirectly clear)
The methods used for such review should explained in a separate paragraph. (How the paper search for the review was done? Which were the criteria for selecting literature?
The section results and discussion are not followed. (It derives from the methods)
Title: I suggest introducing also in the title some reference to the Mediterranean context, which is often commented during the text.
Also, in the results part, the choice of Mediterranean species is not well evident we suggest being more specific in choosing Mediterranean species.
Line 44 The Names of Woody Plants” by Naktuinbouw is the most common reference by the agronomists but not from plant taxonomists. I suggest mention it is referred to such approach.
Line 59 “In the Mediterranean regions, water stress and poor water quality (high salt content) are among the main problems hindering the use of ornamental plants”. There are no references, and we suggest adding them and explaining why. Probably this problem can vary from one city to other cities.
Line 177-78 In ornamental plants, the aesthetic value, calculated as the percentage of necrotic areas in leaves [42], is affected by the increase in EC. It is unclear. The aesthetic value arises from several parameters....
Line 295 The figure 1 is not so relevant, since it summaries an already described physiological mechanism, which is not in the main focus of the paper (in general references to the textbook or paper from which it is derived would be necessary)
Line 393 here I suggest adding a reference on the value of nature in the city: e.g. Capotorti, G., Bonacquisti, S., Abis, L., Aloisi, I., Attorre, F., Bacaro, G., ... & Blasi, C. (2020). More nature in the city. Plant Biosystems-An International Journal Dealing with all Aspects of Plant Biology, 154(6), 1003-1006.
Line 415-419 Fig. 3 “Miller [86] proposed a species selection scheme that included site (i.e., environmental and cultural constraints), social factors (i.e. aesthetics, functions and disruptions) and economic factors (i.e. planting and maintenance costs). Roloff et al. [87] focused on drought tolerance and cold hardiness as critical for the future survival of trees in a changing climate, based on the climatic conditions of the species' places of origin.”
Another extremely important aspect in plant species selection is the ecological aspect (it is partially taken into consideration in the section environment adaptation, but it is not completely approached considering the list of parameters. We suggest considering ecology as of primary importance in species selection; i.e. the equilibrium in the new habitat, avoiding competition with natural species, i.e. avoiding the risk of alien species in the invasions of natural habitats (e.g. Ailanthus altissima can grow in a wide range of conditions, but is must be discouraged; Carpobrotus ssp. can have aesthetic appeal, but it must be discouraged,,,,,,and it can occur for several other species.
Line 576 the theme of antagonistic species should be more clearly explained with examples of species that cannot grow together.
Line 584 Again, the table 1 turns out to be a general scheme found in ecological books, and it is unnecessary.
Author Response
Reviewer 2
Dear reviewer,
The authors would like to thank you for your comments. The manuscript has been accordingly revised. Corrections and suggestions have been implemented in the current version of the manuscript. All the modifications are highlighted in yellow in the manuscript. We hereby provide a point-by-point answer.
The authors
General comments
The article wants to investigate, in a review form, the different stresses of plants in urban environment, such as the mechanisms of tolerance or avoiding stresses by different plant species. The paper also stresses possible agronomic strategies, bio stimulants, mulching, and plant combination which can provide a useful solution to counteract the abiotic stress in the urban environment.
Overall, the article is interesting, summarizing relevant solutions, which can be applied for improving tolerance, however, some relevant changes will improve the interest and originality of the paper.
Author Answer (A.A.): thank you very much for the positive comments and suggestions
In particular, the first part (in particular Paragraph 3 and all the following subsections until paragraph 4.1) should be highly reduced since many of such information can be found in general texts of plant ecology, and it will much more coherent with the title.
A.A.: thank you for the comments. These sections have been shortened considering the comments of the three reviewers and summarized the part paragraphs and sections. We agree that some statements are typically of plant ecology, because our aim was to extend the plant ecology knowledge to practical application and in particular on ornamental sector for plants to be used in urban green area. We apologize if this concept has not been transferred. We revised all the manuscript keeping the focus on abiotic stress, plant behavior in urban environment with the aim to preserve ornamental quality.
The structure doesn’t follow the typical structure of a research or review paper: the aims are not declared (they are indirectly clear)
A.A.: we try to clearer the aims of paper. In particular we added at the end of introduction:
In the present review, after a brief introduction, we analyzed the scientific literature related to woody plants, in particular shrubs, and tolerance to different abiotic stresses, commonly present in urban area, to individuate biological and agronomic strategies to improve this tolerance. We consider the following stresses: drought, salt, hypoxia, shadow, high and low temperatures. We also discuss the sustainable strategy to improve this tolerance: i) biological and hence, the correct choice of genotype (species and/or cultivar) and ii) agronomic tools and management plans (use of biostimulant and arbuscular mycorrhizas, mulching, and association among different species to improve symbiotic effects). The aim of the review was to examine current knowledge to identify what can be done to increase the compatibility of the urban environment with the presence of ornamental plants within green infrastructures, which represent a solution to significantly improve the quality of our cities, which is often compromised from anthropic action, also in view of climate change.
The methods used for such review should explained in a separate paragraph. (How the paper search for the review was done? Which were the criteria for selecting literature?
A.A.: the methods used are added. In particular, we added a specific paragraph on this aspect intitled to ‘Methodology and Literature Research’. As requested also by the reviewer 3, a specific paragraph has been reported indicating how the scientific papers have been selected for the review preparation. The most important scientific databases were consulted for the selection of suitable research papers.
The section results and discussion are not followed (It derives from the methods)
A.A.: since the work is a review there is no separation between results and discussion. The review was organized by topic and in each of it the most relevant findings have been reported and critical elaboration has been carried out, suggesting where appropriate the further investigations.
Title: I suggest introducing also in the title some reference to the Mediterranean context, which is often commented during the text.
A.A.: Thanks for the suggestion; the title is modified in: New strategies to increase the abiotic stress tolerance in woody ornamental plants in Mediterranean climate.
Also, in the results part, the choice of Mediterranean species is not well evident we suggest being more specific in choosing Mediterranean species.
A.A.: We have taken into account that, from a horticultural point of view, plants used in Mediterranean green spaces are those adaptable to this environment, often of exotic origin (their actual presence is often higher than 90%). While acknowledging that the contribution of native plants must be increased, we could not fail to mention other exotic (when not invasive) plants.
Line 44 The Names of Woody Plants” by Naktuinbouw is the most common reference by the agronomists but not from plant taxonomists. I suggest mention it is referred to such approach.
A.A.: Thanks for the comment and suggestion. Our approach to the question was horticultural and agronomic (with broad reference to cultivars that have no meaning for taxonomists); so, the reference in question became the most pertinent. The mention to this characteristic has added to the manuscript.
Line 59 “In the Mediterranean regions, water stress and poor water quality (high salt content) are among the main problems hindering the use of ornamental plants”. There are no references, and we suggest adding them and explaining why. Probably this problem can vary from one city to other cities.
A.A.: Thank you for the comment since we have largely worked on this topic a specific reference has been added: Toscano, S., Ferrante, A., & Romano, D. (2019). Response of Mediterranean ornamental plants to drought stress. Horticulturae, 5(1), 6.
Line 177-78 In ornamental plants, the aesthetic value, calculated as the percentage of necrotic areas in leaves [42], is affected by the increase in EC. It is unclear. The aesthetic value arises from several parameters ...
A.A.: The aesthetic value of a plant is connected to numerous parameters (habit, characteristics of the foliage and flowers number and turnover, etc.); your observation made us realize that the sentence was convoluted; what we meant was: the increase of EC modify the morphological and physiological characteristics of plants and in particular increase the percentage of leaf necrosis, reducing the aesthetic value that is the key parameter of assessment of ornamental plants.
Line 295 The figure 1 is not so relevant, since it summaries an already described physiological mechanism, which is not in the main focus of the paper (in general references to the textbook or paper from which it is derived would be necessary)
A.A.: we agree that is a well-known cycle; therefore, the figure 1 has been moved to the Supplementary Materials
Line 393 here I suggest adding a reference on the value of nature in the city: e.g. Capotorti, G., Bonacquisti, S., Abis, L., Aloisi, I., Attorre, F., Bacaro, G., ... & Blasi, C. (2020). More nature in the city. Plant Biosystems-An International Journal Dealing with all Aspects of Plant Biology, 154(6), 1003-1006.
A.A.: Thanks for the suggestion. The reference was added.
Line 415-419 Fig. 3 “Miller [86] proposed a species selection scheme that included site (i.e., environmental and cultural constraints), social factors (i.e. aesthetics, functions and disruptions) and economic factors (i.e. planting and maintenance costs). Roloff et al. [87] focused on drought tolerance and cold hardiness as critical for the future survival of trees in a changing climate, based on the climatic conditions of the species' places of origin.”
Another extremely important aspect in plant species selection is the ecological aspect (it is partially taken into consideration in the section environment adaptation, but it is not completely approached considering the list of parameters. We suggest considering ecology as of primary importance in species selection; i.e. the equilibrium in the new habitat, avoiding competition with natural species, i.e. avoiding the risk of alien species in the invasions of natural habitats (e.g. Ailanthus altissima can grow in a wide range of conditions, but is must be discouraged; Carpobrotus ssp. can have aesthetic appeal, but it must be discouraged and it can occur for several other species.
A.A.: Ecological aspects must be taken into account since they can strongly affect the quality determinants in ornamental plants. However, these take on relevance according to the size of the green infrastructures. We cannot forget that when space is parceled out (for example, road trees, traffic reservations), we are faced with an environment that is more similar to a cultivated space than a natural ecosystem. We are aware of the risks of alien species, but the references in the review are above all in relation to spaces cultivated by man for ornamental and fruition purposes within the city.
Line 576 The theme of antagonistic species should be more clearly explained with examples of species that cannot grow together.
A.A.: The antagonist effects are well-known in vegetable cultivations, while less information is available for ornamental species. However, some examples of allelopathic effects have been added and the lack of information has been highlighted in the text.
Line 584 Again, the table 1 turns out to be a general scheme found in ecological books, and it is unnecessary.
A.A.: As suggested by the reviewer, the table has been removed.
Reviewer 3 Report
The manuscript has illustrated new strategies and techniques to increase the abiotic stress tolerance in woody ornamental plants.
*In the Keywords, please replace ( , ) with ( ; ) after green areas.
*In the section one which is INTRODUCTION, authors have used many paragraphs. But, each paragraph should start with new topic and information.
*Please, write about METHODOLOGY of research at the last paragraph of Introduction, the method and sources of checking and keywords which have been used in this manuscript, or make a new paragraph and section as SECTION TWO, and write about it.
*The section TWO (The ornamental shrubby plants in the urban environment) is not enough, and authors need to write more about this topic.
*Authors need to design one Table as well as what they have illustrated in the lyrics to describe and explain the impacts of different kinds of stresses both biotic and abiotic stresses on ornamental crops with Referencing.
*Again in section (3.1) which is about Drought Stress, paragraphing is too much and authors should design it and re-arrange it again.
*I suggest for each section, authors design one Table, for example one table for Hypoxia, and one table for Pollution, etc.
*Line 469, ///Plant biostimulant/ , why do authors use one time /// and another time / ????
*Conclusion part is not enough, and it should be re-written, the main point of Conclusion is not clear, please revise it.
*The format of all References should be double-checked, for example, you can check references number 24 and 25, Reference number 28, number 45,
*I suggest to use more updated References for this article, for example published article about this interesting subject from 2015 to 2023.
I suggest that authors use English service of MDPI, or ask one native English to revise the article.
Author Response
Reviewer 3
Comments and Suggestions for Authors
The manuscript has illustrated new strategies and techniques to increase the abiotic stress tolerance in woody ornamental plants.
*In the Keywords, please replace ( , ) with ( ; ) after green areas.
Author Answer (A.A.): Done
*In the section one which is INTRODUCTION, authors have used many paragraphs. But, each paragraph should start with new topic and information.
A.A.: Thanks for the suggestion. The introduction was modified
*Please, write about METHODOLOGY of research at the last paragraph of Introduction, the method and sources of checking and keywords which have been used in this manuscript, or make a new paragraph and section as SECTION TWO, and write about it.
A.A.: Thanks for the suggestion. We are added a specific paragraph on this aspect devoted to ‘Methodology and Literature Research’. As requested also by the reviewer 2, a specific paragraph has been added, indicating how the scientific papers have been selected for the review preparation. The most important scientific databases were consulted for the selection of suitable research papers.
*The section TWO (The ornamental shrubby plants in the urban environment) is not enough, and authors need to write more about this topic.
A.A.: This paragraph has been extended and modified as requested by the other reviewer.
*Authors need to design one Table as well as what they have illustrated in the lyrics to describe and explain the impacts of different kinds of stresses both biotic and abiotic stresses on ornamental crops with Referencing.
A.A.: As requested by the reviewer a table has been added to the text highlighting the stress effects on ornamental parameters and the tolerance traits that should be used for selecting tolerant species.
*Again in section (3.1) which is about Drought Stress, paragraphing is too much and authors should design it and re-arrange it again.
A.A.: The section has been completely revised and shortened.
*I suggest for each section, authors design one Table, for example one table for Hypoxia, and one table for Pollution, etc.
A.A. For each section a specific table has been added.
*Line 469, ///Plant biostimulant/, why do authors use one time /// and another time / ????
A.A. The text has been revised and terms have been constantly used through the text.
*Conclusion part is not enough, and it should be re-written, the main point of Conclusion is not clear, please revise it.
A.A.: the conclusions have been revised and re-written considering the most relevant information coming out form the literature review.
*The format of all References should be double-checked, for example, you can check references number 24 and 25, Reference number 28, number 45,
A.A. The references have been formatted and reported following the journal guidelines.
*I suggest to use more updated References for this article, for example published article about this interesting subject from 2015 to 2023.
A.A. The references have been selected and cited considering the robustness of data quality and elaboration, the importance and relevance of to our review work, and finally the preference has been also considered by considering the most recent and relevant works that merited to be included.
Comments on the Quality of English Language: I suggest that authors use English service of MDPI, or ask one native English to revise the article.
A.A. The English editing has been performed using IA software such as Grammarly pro.
Round 2
Reviewer 2 Report
the paper has been improved according to the reviewer suggestions and now is ready for publication
Author Response
Response to Reviewer 2 Comments
Comments and Suggestions for Authors
The paper has been improved according to the reviewer suggestions and now is ready for publication
Author Answer (A.A.): Thank you very much for your positive evaluation.
Reviewer 3 Report
The article can be accepted in present format.
Author Response
Response to Reviewer 3 Comments
Comments and Suggestions for Authors
The article can be accepted in present format.
Author Answer (A.A.): Thank you very much for your positive evaluation.